# Cardiac Tyrosine 97 Phosphorylation of Cytochrome *c* Regulates Respiration and Apoptosis

**DOI:** 10.3390/ijms26031314

**Published:** 2025-02-04

**Authors:** Paul T. Morse, Vignesh Pasupathi, Susanna Vuljaj, Nabil Yazdi, Matthew P. Zurek, Junmei Wan, Icksoo Lee, Asmita Vaishnav, Brian F.P. Edwards, Tasnim Arroum, Maik Hüttemann

**Affiliations:** 1Center for Molecular Medicine and Genetics, Wayne State University, Detroit, MI 48201, USA; morsepa@wayne.edu (P.T.M.);; 2Department of Biochemistry, Microbiology, and Immunology, Wayne State University, Detroit, MI 48201, USA; 3College of Medicine, Dankook University, Cheonan-si 31116, Chungcheongnam-do, Republic of Korea

**Keywords:** cytochrome *c*, mitochondrial respiration, apoptosis: reactive oxygen species, myocardial infarction, ischemia-reperfusion

## Abstract

It was previously reported that tyrosine 97 (Y97) of cytochrome *c* is phosphorylated in cow heart tissue under physiological conditions. Y97 phosphorylation was shown to partially inhibit respiration in vitro in the reaction with purified cytochrome *c* oxidase. Here, we use phosphomimetic Y97E Cyt*c* to further characterize the functional effects of this modification both in vitro and in cell culture models. In vitro, phosphomimetic Y97E Cyt*c* showed lower activity in the reaction with purified cow heart cytochrome *c* oxidase (COX), decreased caspase-3 activity, and reduced rate of reduction. Additionally, the phosphomimetic Y97E Cyt*c* tended to be resistant to heme degradation and showed an increased rate of oxidation. Intact mouse Cyt*c* double knockout fibroblasts were transfected with plasmids coding for phosphomimetic Y97E Cyt*c* and other variants. Compared to cells expressing wild-type Cyt*c*, the cells expressing phosphomimetic Y97E Cyt*c* showed reduced respiration, mitochondrial membrane potential, and reactive oxygen species production, and protection from apoptosis. In an oxygen–glucose deprivation/reoxygenation cell culture model of ischemia/reperfusion injury, mitochondrial membrane potential and reactive oxygen species production were decreased. These data show that Cyt*c* phosphorylation controls the overall flux through the electron transport chain by maintaining optimal intermediate ΔΨm potentials for efficient ATP production while minimizing reactive oxygen species production, thus protecting the cell from apoptosis.

## 1. Introduction

Cytochrome *c* (Cyt*c*) is a small globular protein with a single covalently linked heme group per peptide chain. Most mitochondria Cyt*c* species contain between 94 and 114 amino acids, depending on the species, and 104 amino acids in mammals [1]. Cyt*c* acts as a molecular switch between cellular life and death. The primary life-sustaining role of Cyt*c* is to shuttle the electrons from complex III to complex IV (cytochrome *c* oxidase; COX), also known as the terminal O_2_ acceptor within the electron transport chain (ETC) [2]. The electron transfer from Cyt*c* to COX is the proposed rate-limiting step of the ETC (reviewed in [3]) and, therefore, is subject to tight regulation. Cyt*c* also acts as a detoxifying agent to dispose of reactive oxygen species (ROS) [4,5] and as part of the Erv1-Mia40 redox-coupled protein import pathway [6,7]. Cyt*c* has even been shown to adopt different conformations depending on the location of the protein within the mitochondria [8]. Under stress conditions, Cyt*c* is released from the mitochondria into the cytosol, acting as an inducer of cell death by initiating intrinsic apoptosis [9,10,11]. During this process, Cyt*c* binds to apoptotic protease activating factor 1 (Apaf-1) to form the apoptosome, activating procaspase-9 to caspase-9, which cleaves procaspase-3 to caspase-3 and commits the cell to apoptosis. However, this is a delicate balance, as the release of sublethal amounts of Cyt*c* has been shown to desensitize cancer cells to apoptosis [12]. Cyt*c* has other pro-apoptotic functions, such as cardiolipin peroxidase activity and ROS formation via reduction of p66^Shc^ [11,13,14,15,16,17,18]. Recent work has also shown that Cyt*c* can translocate to the nucleus as part of the DNA damage response [19,20]. Altogether, these diverse functions of Cyt*c* require tight regulatory control by the cell, such as allosteric regulation via ATP binding, expression of tissue-specific isoforms (somatic and testis), and post-translational modifications (PTMs) (reviewed in [21,22]).

Many PTMs of Cyt*c* have been studied, including acetylation, carbonylation [23,24,25], deamidation [26], glycation [27,28,29,30], glycosylation [31,32], homocysteinylation [33,34,35,36], nitration [37,38,39,40,41], nitrosylation [42], phosphorylation, and sulfoxidation [43,44,45,46]. To date, six tissue-specific phosphorylations of Cyt*c* have been mapped by mass spectrometric analysis (reviewed in [21]): T28 and T58 in the kidney, S47 in the brain, Y48 in the liver, and T49 and Y97 in the heart. Additionally, there are two characterized disease-specific acetylations of Cyt*c* (reviewed in [21]) K39 in ischemic skeletal muscle and K53 in prostate cancer. We and others have characterized the functional effects of the phosphorylations directly and by using in vitro and in-cell culture models with phosphomimetics. Typically, phosphorylations of Cyt*c* and their respective phosphomimetics demonstrate reduced respiration, mitochondrial membrane potential (ΔΨ_m_), ROS production, and apoptosis. Recently, Cyt*c* phosphorylation has been shown to modify the newly discovered nuclear functions of Cyt*c* [47]. Additionally, Cyt*c* phosphorylation can be lost under ischemia-reperfusion (I/R) injury, likely due to activation of Ca^+^-sensitive phosphatases, such as S47 dephosphorylation in ischemic brain [48,49], which promotes cell death. We propose that Cyt*c* phosphorylation maintains optimal intermediate ΔΨ_m_ levels for efficient ATP production but limited ROS generation while dephosphorylation induces pathologically high ΔΨ_m_ levels [7,8,9,10,13] and excessive ROS production, leading to cell death.

We previously identified phosphorylation of Cyt*c* on Y97 in the cow heart under basal conditions using our specialized purification protocol, which preserves the physiological phosphorylation status [48]. Phosphorylated Cyt*c* Y97 demonstrated sigmoidal kinetics in the reaction with purified cow COX, while the dephosphorylated protein produced a hyperbolic response [50]. Additionally, phosphorylation at Y97 shifted the heme iron-methionine 80 absorption peak from 695 nm to 687 nm, indicating perturbations close to the catalytic heme group. Other studies using a phosphomimetic system demonstrated reduced thermal stability [51] and caspase-3 activity [52]. We also reported that neuroprotective insulin treatment induced this modification in the brain, resulting in reduced Cyt*c* release from the mitochondria and reduced overall neuronal death after an ischemic insult [53].

In the present study, we aim to expand our understanding of the effects of Cyt*c* Y97 phosphorylation on multiple Cyt*c* functions by using mutagenesis with purified Cyt*c* variants and with cell lines stably expressing Cyt*c* variants, including WT, phosphomimetic mutant Y97E, and phenylalanyl mutant Y97F as an additional control. In vitro, we found that phosphomimetic Y97E substitution causes inhibition of respiration in the reaction with COX, similar to what is observed with the purified Y97 phosphorylated Cyt*c*, and reduced caspase-3 activity. In the cell culture model, we found that by introducing plasmids coding for phosphomimetic Y97E and WT Cyt*c* into Cyt*c* double knockout cells, phosphomimetic substitution reduces intact cell respiration, ΔΨ_m_, and ROS production. Additionally, in an oxygen–glucose deprivation/reoxygenation (OGD/R) model, which mimics I/R injury, Y97E is protective against cell death by maintaining an optimal ΔΨ_m_ and reduced levels of ROS. These data suggest that phosphorylation of Y97 regulates the overall ETC flux under normal conditions, maintains a healthy intermediate ΔΨ_m_, and controls mitochondrial respiration, whereas under stress conditions such as I/R injury, dephosphorylated Cyt*c* drives maximal ETC flux, thus causing overproduction of ROS and cell death.

## 2. Results

### 2.1. Overexpression and Purification of Functional Cytochrome c Variants in Bacterial Cells

Y97 phosphorylation was studied in vitro via mutagenesis to create phosphomimetic Cyt*c* using an *E. coli* overexpression system. We have used the approach by replacing the rodent (somatic mouse and rat Cyt*c* have the same sequence) wild-type (WT) Y97 Cyt*c* residue (Figure 1A) with the negatively charged amino acid glutamate (Y97E), which mimics the in vivo phosphorylated Y97 due to glutamate and phospho-tyrosine both possessing a negative charge. Using glutamate as a mimetic for phospho-tyrosine is an established technique for both Cyt*c* [21] and other proteins in general [54]. We also generated additional control Y97 phenylalanine (Y97F), which cannot be phosphorylated. Using an *E. coli* overexpression system and our purification protocol, we generated significant amounts of the recombinant proteins. The Coomassie blue stained gel (Figure 1B) and reduced UV–Vis spectra (Figure 1C) show that isolated Cyt*c* variants were highly pure and demonstrated correct folding due to the presence of the characteristic α, β, and γ peaks. In the previous publication where Y97 Cyt*c* phosphorylation was initially discovered, in vivo, partially Y97 phosphorylated Cyt*c* demonstrated a blue shift of the 695 nm peak to 687 nm. Here, phosphomimetic Y97E Cyt*c* also demonstrates a shift and broadening of the 695 nm peak to 687 nm (Figure 1C inset), indicating that Y97E replacement is a good mimetic for in vivo Y97 phosphorylation

### 2.2. Y97E Phosphomimetic Cytc Shows Decreased COX Activity

In the previous publication where Y97 Cyt*c* phosphorylation was initially discovered, in vivo partially Y97 phosphorylated Cyt*c* purified from bovine heart demonstrated sigmoidal kinetics with a maximal turnover of 32 s^−1^ and a K_m_ of 5.5 μM, compared to the un-phosphorylated Cyt*c* showing a hyperbolic response with a maximal turnover of 32 s^−1^ and a K_m_ of 2.5 μM [50]. Here, COX activity was measured with the recombinant Cyt*c* variants in reaction with bovine heart COX (Figure 1D). Similarly to the data previously seen with Y97 phosphorylated Cyt*c*, phosphomimetic Y97E Cyt*c* showed pronounced sigmoidal kinetics and a maximal turnover number of 7.0 s^−1^. In contrast, WT Cyt*c* and Y97F Cyt*c* show hyperbolic responses with a maximum turnover of 9.3 s^−1^ and 9.2 s^−1^, respectively. The K_m_ for Y97E Cyt*c* in the reaction with COX was 9.1 μM, while the K_m_ for WT Cyt*c* and Y97F Cyt*c* were 4.3 μM and 4.8 μM, respectively. While the absolute values for V_max_ and K_m_ seen here differ from those of the previous publication, likely due to differences between using bovine Cyt*c* (previous publication) and rodent Cyt*c* (here) with bovine COX, the trends are overall similar. These data suggest that Y97E replacement is a good mimetic for in vivo Y97 phosphorylation, showing sigmoidal kinetics compared to WT.

### 2.3. Y97E Phosphomimetic Cytc Shows Reduced Caspase-3 Activity

Cyt*c* plays a central role in I/R injury via Apaf-1 interaction and caspase-3 activation. To test the effect of Y97 phosphorylation on apoptosis, where Cyt*c* interacts with Apaf-1 to form the apoptosome, we analyzed Cyt*c* variants in a cell-free caspase-3 assay. Phosphomimetic Y97E Cyt*c* showed 64% decreased caspase-3 activity compared to the WT (Figure 1E).

### 2.4. Phosphomimetic Y97E Cytc Displays a Higher Oxidation Rate, Lower Reduction Rate, and Is Partially Resistant to Heme Degradation by Hydrogen Peroxide

Cyt*c* plays an important role as a ROS scavenger. To study this behavior, we analyzed the rate of oxidation of ferro Cyt*c* variants in the presence of 100 µM of H_2_O_2_ and the rate of reduction of ferri Cyt*c* variants in the presence of superoxide generated by a hypoxanthine/xanthine oxidase reaction system. Phosphomimetic Y97E Cyt*c* displayed a 36% higher oxidation rate (Figure 1F) and a 42% lower reduction rate, both compared to the WT (Figure 1G). For the rate of reduction, an additional group was included as a negative control which contained superoxide dismutase (SOD), detoxifying superoxide generated by the hypoxanthine/xanthine oxidase system. While Cyt*c* plays an important role in ROS scavenging, Cyt*c* can lose its functionality at high ROS concentrations via the destruction of the heme group. To assess the stability of the heme group, we spectrophotometrically tracked the degradation of the heme catalytic site of Cyt*c* over time upon the addition of 3 mM H_2_O_2_. Loss of absorbance of the Soret peak of Cyt*c* at 408 nm is characteristic of heme degradation. Phosphomimetic Y97E Cyt*c* did not show any statistically significant difference in heme degradation at 800 s compared to the WT. Interestingly, Y97F Cyt*c* did show increased, but not statistically significant, heme degradation, implying that a nonpolar residue at the site destabilizes the heme group (Figure 1H).

### 2.5. Mutagenesis and Expression of Cytochrome c Variants in Stable Cell Line

To study the functional effects of a Y97E phosphomimetic substitution of Cyt*c* in an intact cell culture system, we created cell lines stably expressing WT Cyt*c*, Y97F Cyt*c*, and Y97E Cyt*c*. Plasmids encoding these Cyt*c* variants were transfected into a Cyt*c* double knockout mouse lung fibroblast cell line, where both the rodent testes and somatic isoforms of Cyt*c* were knocked out [55]. Cell lines that were equally expressing WT, Y97F, and Y97E Cyt*c,* along with a Cyt*c*-null empty vector (EV) control, were used for the functional studies. Western blot analysis showed that Cyt*c* variants were expressed at similar levels in the cell lines (Figure 2A).

### 2.6. Mitochondrial Respiration Is Inhibited in Cells Expressing Phosphomimetic Y97E Cytc

Glycolysis and mitochondrial oxidative phosphorylation are two primary energy metabolism pathways that support all other cellular functions. The reaction between Cyt*c* and COX has been proposed to be the rate-limiting step of the ETC (reviewed in [3]), and therefore, alterations of Cyt*c* may impact overall cellular respiration. Intact cellular respiration of the cells expressing the Cyt*c* variants was measured via oxygen consumption rate (OCR) during a mitochondrial stress test using a Seahorse bioanalyzer (Figure 2B). The cells expressing phosphomimetic Y97E Cyt*c* displayed a reduced basal respiration rate of 46% compared to the cells expressing the WT (Figure 2C). Additionally, cells expressing phosphomimetic Y97E Cyt*c* showed decreased ATP-coupled respiration, maximal respiration, and spare respiratory capacity compared to cells expressing the WT (Appendix A).

### 2.7. Mitochondrial Membrane Potential and ROS Production Are Decreased in Cells Expressing Phosphomimetic Y97E Cytc

Since the respiration rate was decreased in cells expressing phosphomimetic Y97E Cyt*c*, we hypothesized that there should be reduced mitochondrial membrane potential (ΔΨ_m_) and, consequently, reduced ROS levels. ΔΨ_m_ was measured with JC-10, a voltage-dependent probe. We found that the red-to-green fluorescence ratio was decreased by 25% in cells expressing phosphomimetic Y97E Cyt*c* compared to cells expressing the WT, indicating a reduction of ΔΨ_m_ (Figure 2D).

Previous research has demonstrated a correlation between ΔΨ_m_ and ROS production [56], which is particularly relevant in conditions of cellular stress, such as I/R injury. As high membrane potentials drive ROS production, we measured ROS levels using the MitoSOX probe. We found that MitoSOX fluorescence was 36% decreased in cells expressing phosphomimetic Y97E Cyt*c* compared to cells expressing the WT, indicating reduced mitochondrial ROS levels (Figure 2E).

### 2.8. Cells Expressing Phosphomimetic Y97E Cytc Are Protected from Cell Death

We analyzed apoptosis induced by H_2_O_2_ or staurosporine treatment for the cell lines expressing the Cyt*c* variants. Necrosis and early and late apoptosis were assessed using annexin V/propidium iodide (PI) staining followed by flow cytometry analysis. H_2_O_2_ and staurosporine are known to induce cell death via oxidative stress [57] or inhibition of protein kinases [58], respectively. The cells were subjected to 400 µM of H_2_O_2_ for 16 h and 1 µM of staurosporine for 5 h, respectively. In both stress-inducing conditions, cells expressing the phosphomimetic Y97E variant showed decreases in total cell death. Treatment with H_2_O_2_ resulted in 36% cell death in cells expressing the phosphomimetic Y97E Cyt*c* compared to 45% cell death in cells expressing the WT (Figure 3A). Similarly, treatment with staurosporine resulted in 16% cell death in cells expressing the phosphomimetic Y97E Cyt*c* compared to 40% cell death in cells expressing the WT (Figure 3B). Overall, the decrease in total cell death for the cells expressing the phosphomimetic Y97E Cyt*c* mimics the decreases seen with the in vitro caspase-3 activity assay, highlighting the role of Y97 phosphorylation in reducing the pro-apoptotic features of Cyt*c*.

### 2.9. Cells Expressing Phosphomimetic Y97E Cytc Suppress ROS Production upon Oxygen–Glucose Deprivation/Reoxygenation (OGD/R)

Mitochondrial dysfunction during reperfusion following ischemia has been linked to increased mitochondrial membrane potential and ROS-induced damage of mitochondrial components [59]. During reperfusion, ΔΨ_m_ hyperpolarizes, which drives the production of ROS. Other phosphorylations of Cyt*c* that are present under basal conditions are lost during ischemia due to the depletion of the kinase substrate ATP and the loss of calcium homeostasis, which results in the activation of mitochondrial phosphatases [48,49]. We employed the oxygen–glucose deprivation/reoxygenation (OGD/R) model to further study this relationship. The cells expressing the Cyt*c* variants were exposed to 1% oxygen in glucose-depleted media for 90 min, mimicking the ischemic condition. This was followed by 30 min of simulated reperfusion in oxygen- and glucose-containing media supplemented with either JC-10 or MitoSOX probe to study ΔΨ_m_ and ROS production. Excluding EV cells, which do not contain Cyt*c* and are thus protected from Cyt*c*-mediated apoptosis induction, cells expressing the phosphomimetic Y97E Cyt*c* showed the lowest ΔΨ_m_ levels after OGD/R compared to cells expressing other Cyt*c* variants (Figure 4A). Similar findings were observed after OGD/R for MitoSOX, i.e., the cells expressing phosphomimetic Y97E Cyt*c* showed the lowest mitochondrial ROS production after OGD/R, excluding the EV (Figure 4B). These results suggest a protective role of Y97E phosphomimetic replacement and, thus, Y97 phosphorylation in I/R injury.

## 3. Discussion

Generally, tissue-specific phosphorylations of Cyt*c* tend to be present under physiological conditions, leading to partial inhibition of oxidative phosphorylation and significant inhibition of apoptosis (reviewed in [21]). These provide slight tweaks to mitochondrial functioning to optimize tissue metabolism and make the tissues resilient to physiological stressors. The loss of these phosphorylations during ischemia, due to rapid depletion of the kinase substrate ATP, in addition to the loss of mitochondrial Ca^2+^ homeostasis leading to activation of phosphatases, results in reperfusion injury due to ΔΨ_m_ hyperpolarization driving increased ETC activity, reverse electron transfer, and ROS bursts (reviewed in [60]). We propose that the loss of the inhibitory phosphorylations during ischemia is a futile attempt of the cell trying to upregulate oxidative phosphorylation and energy production, with another deleterious effect of sensitizing the tissue to apoptosis [48,49]. Given that Y97 phosphorylation of Cyt*c* was discovered under physiological conditions in the mammalian heart, an organ with known vulnerability to I/R injury resulting from myocardial infarction, it is highly likely that loss of this phosphorylation contributes to I/R injury in a way similar to that seen of S47 phosphorylation of Cyt*c* in the brain [48].

Our lab previously showed that Y97 phosphorylation of Cyt*c* is an important modification that regulates COX enzyme activity in the mammalian heart [50]. Here, we expand on that previous work by characterizing phosphomimetic Y97E Cyt*c* using both purified protein and transfected cell culture models. Previous work on Cyt*c* has shown that glutamate is a suitable mimic for phospho-tyrosine [51]. Y97 is conserved in mammals, with only a few species of plants or microorganisms expressing a different residue at the site, most commonly F97 [1]. One of the two main functions of Cyt*c* is as a single electron carrier from complex III to COX in the ETC, which is the proposed rate-limiting step of the ETC under physiological conditions [3]. The publication that initially discovered phosphorylated Y97 Cyt*c*, which was purified from bovine heart, showed that the phosphorylation resulted in sigmoidal kinetics in reaction with COX, whereas the dephosphorylated protein showed a hyperbolic response [50]. Our data here with the phosphomimetic Y97E Cyt*c* agree with these results, with the phosphomimetic Y97E Cyt*c* showing sigmoidal kinetics and the WT, non-phosphorylated protein showing a hyperbolic response. Furthermore, the UV–Vis spectra of phosphomimetic Y97E show the characteristic blue shift of the typical 695 nm peak to 687 nm, similar to the in vivo Y97 phosphorylated Cyt*c*. These results further support the notion that Y97E replacement mimics the effects seen with Y97 phosphorylated Cyt*c*.

The second main function of Cyt*c* is as a key pro-apoptotic signal [61]. Upon its release from the mitochondria to the cytosol, Cyt*c* interacts with Apaf-1 to form the apoptosome. This results in the activation of caspase-9 and triggers the downstream caspase cascade. Our data here shows statistically significant reductions in caspase-3 activity, with a 64% reduction in caspase-3 activity when using phosphomimetic Y97E Cyt*c* compared to WT. Interestingly, when analyzed with a Swiss PDB viewer using a published docking model of Cyt*c* and Apaf-1 (3JBT.pdb), Y97 is not part of the Apaf-1 interaction site [62]. It is spatially located closest to Apaf-1 residues E659 and D1106 with an approximate distance of 14 Å (Figure 5). The binding between Cyt*c* and Apaf-1 is known to be driven by electrostatic interactions, with the positively charged Cyt*c* binding to its negatively charged binding pocket on Apaf-1 [62]. The gain of a negative charge with phosphomimetic Y97E Cyt*c* may cause globally decreased binding to Apaf-1. Additionally, when looking at purified phosphomimetic Y97E Cyt*c* protein, we identified an increased rate of oxidation with H_2_O_2_ and a decreased rate of reduction with superoxide.

Furthermore, we characterized phosphomimetic Y97E Cyt*c* in a cell culture model. Rodents possess two active Cyt*c* isoforms, somatic and testes, while humans possess a single active Cyt*c* isoform that possesses features of both rodent isoforms. Knocking out the rodent somatic isoform induces expression of the testes isoform to ensure metabolic competence [55]. Therefore, we employed a double knockout model where both the somatic and testes isoforms are knocked out. These double knockout cells were transfected with the recombinant Cyt*c* variants to mimic the effects of Y97 phosphorylation. Our results show decreased respiration, decreased ΔΨ_m_, and decreased ROS production in cells expressing the phosphomimetic Y97E Cyt*c*, highlighting the relationship between ΔΨ_m_, respiration, and ROS production highlighted above (Figure 6). When looking at apoptosis triggered by H_2_O_2_ and staurosporine, the cells expressing phosphomimetic Y97E Cyt*c* showed decreased total cell death, which matches the data observed using the recombinant phosphomimetic Y97E Cyt*c* protein, which demonstrated an increased rate of oxidation with H_2_O_2_ and decreased caspase-3 activity.

The partial inhibition of respiration observed in intact cells expressing the phosphomimetic Y97E Cyt*c* is significant because of the relationship between respiration, ΔΨ_m_, and ROS production. It is known that 95% of ROS are generated from electron leaks in the respiratory complexes I and III [63]. Under normal conditions, maintenance of intermediate ΔΨ_m_ levels (80–140 mV) avoids ROS generation but provides the full capability to produce ATP because maximal rates of ATP synthesis by ATP synthase take place at an intermediate ΔΨ_m_ of 80–140 mV [64]. At ΔΨ_m_ exceeding 140 mV, ROS production at ETC complexes I and III increases exponentially, leading to cell death [65]. This relationship between ΔΨ_m_ and ROS is further supported by the results presented here under basal conditions and using the OGD/R model, which simulates I/R injury. In the OGD/R model, cells expressing unphosphorylated WT Cyt*c* showed increased ΔΨ_m_, which triggered a similar increase in ROS production. The cells expressing the phosphomimetic Y97E Cyt*c* demonstrated lower ΔΨ_m,_ which attenuated ROS production. However, on the other hand, phosphomimetic Y97E Cyt*c* protein showed a lower rate of reduction by superoxide. It is possible that this may be partially due to the repulsion between the negative charges of the phosphomimetic glutamate and superoxide. An alternate explanation provided by previous research is that point mutations at Y97 lead to subtle changes in the hydrogen bonding network surrounding the heme group, leading to perturbed redox properties under higher electrical fields [66].

The secondary control, unphosphorylated Y97F, often displayed an intermediate phenotype compared to the WT and the phosphomimetic Y97E Cyt*c*, such as the results seen for the caspase-3 activity, rate of oxidation, and rate of reduction assays. This highlights the regulatory importance of the tyrosine residue at site 97, which allows for phenotypic shifting depending on the phosphorylation state. However, phenylalanine mutation is still tolerated at site 97 and is seen in many species of plants [1]. This provides a possible evolutionary reason for the intermediate phenotype: plants lack the regulatory capability of the phosphorylation site but instead possess a mutation that shows features of both the phosphorylated and unphosphorylated Cyt*c* seen in mammals. Surprisingly, cells expressing Y97F Cyt*c* tend to behave either like cells expressing the WT or the Y97E Cyt*c*, depending on the assay. For the basal respiration and cell death assays, cells expressing Y97F Cyt*c* tend to behave like cells expressing the WT Cyt*c*. On the other hand, for the ΔΨ_m_ and mitochondrial ROS production assays, cells expressing Y97F Cyt*c* tend to behave more like the cells expressing the Y97E Cyt*c*. This highlights the complex relationship between mitochondrial respiration, ΔΨ_m_, and mitochondrial ROS production.

A previous publication also looked at the WT, Y97F, and Y97E Cyt*c* variants [51]. In that publication, García-Heredia et al. found that the midpoint redox potential and cardiolipin peroxidase activity of phosphomimetic Y97E Cyt*c* were not different compared to the WT. Interestingly, the authors did find that the phosphomimetic Y97E Cyt*c* showed dramatically decreased thermal stability, with the melting temperature dropping by 30 °C compared to the WT. However, the authors attributed this not to the effect of the negative charge but rather to the loss of the aromatic ring, as Y97F Cyt*c* was comparable to the WT.

Another publication studied Y97 Cyt*c* phosphorylation by substituting Y97 with the noncanonical amino acid *p*-carboxymethyl-L-phenylalanine (pCMF) [52]. This study reported increased COX activity using phosphomimetic Y97pCMF Cyt*c*, which contradicts data previously obtained with Y97 phosphorylated Cyt*c*—i.e., the gold standard—and data presented here on phosphomimetic Y97E Cyt*c*, which displays altered COX kinetics and slight enzymatic inhibition, matching those observed with the in vivo phosphorylated protein. The authors proposed that this stimulation was due to a double regulatory mechanism via the phosphorylation and increased supercomplex formation, which the authors observed in a yeast model. They also reported reduced ROS production and caspase-3 activity using phosphomimetic Y97pCMF Cyt*c*, although ROS scavenging activity and cardiolipin binding were unchanged.

Interestingly, another study identified that Y97 phosphorylation of Cyt*c* could be induced in ischemic pig and rat brains using insulin [53]. Insulin-induced Y97 phosphorylation of Cyt*c* resulted in significant neuroprotection, with a 49% increase in NeuN-positive neurons compared to the ischemia condition alone.

In conclusion, our results suggest that Cyt*c* Y97 phosphorylation results in optimal, intermediate membrane potential levels, lower ROS production, and significant protection from cell death upon OGD/R (Figure 6). Therefore, Cyt*c* Y97 phosphorylation is a beneficial modification in cardiac muscle under physiological conditions, but its loss during ischemia sensitizes cardiac tissue to reperfusion injury. Finally, recent research studying posttranslational modifications of Cyt*c* supports the concept that such small chemical modifications of this evolutionarily highly optimized protein control electron flux in the ETC, putting it in a central position of regulating ΔΨ_m_, energy production and ROS in a highly fine-tuned tissue-specific manner.

## 4. Materials and Methods

All chemicals and reagents were purchased from MilliporeSigma (Burlington, MA, USA) unless otherwise specified.

### 4.1. Bacterial Overexpression and Recombinant Protein Purification

The codon corresponding to Y97 of the somatic rodent Cyt*c* (WT) cDNA cloned into the pLW01 expression vector was mutated to a glutamate residue (Y97E) as a phosphomimetic replacement or phenylalanine as an additional control (Y97F) which cannot be phosphorylated using the quick-change site-directed mutagenesis kit (Agilent Technologies, Santa Clara, CA, USA) as directed by the manufacturer. The pLW01 plasmid also contained the sequence for *S. cerevisiae* heme lyase, which bacteria lack, in order to successfully covalently attach the heme group to Cyt*c* [67]. The following primers were used to mutate Y97 of the Cyt*c* sequence: Y97E forward primer 5′-GACCTAATAGCT**GAG**CTTAAAAAGGC-3′, Y97E reverse primer 5′-GCCTTTTTAAG**CTC**AGCTATTAGGTC-3′, Y97F forward primer 5′-GGCAGACCTAATAGCT**TTC**CTTAAAAAGGC-3′, and Y97F reverse primer 5′-GCCTTTTTAAG**GAA**AGCTATTAGGTCTGCC-3′. Using the Dpn I restriction enzyme, the parental supercoiled dsDNA was digested for 1 h at 37 °C. The resulting PCR products were transformed into XL 10-Gold Ultracompetent cells. The plasmids from the overnight culture were purified from individual colonies using a QIAprep spin column miniprep kit (#27106, Qiagen; Valencia, CA, USA). Plasmid purity and quantity were assessed using a Nanodrop 1000 spectrophotometer, and the desired mutations were confirmed by sequencing.

The plasmids were transformed into chemically competent *Escherichia coli* strain, C41(DE3) cells (Biosearch technologies; Tucson, AZ, USA) according to the manufacturer’s protocol in order to overexpress the Cyt*c* variants. Selected clones were inoculated into 10 mL of terrific broth containing 0.1 mg/mL of carbenicillin and grown overnight at 37 °C under 225 to 250 RPM agitation. The overnight culture was inoculated into several flasks containing 5 L of carbenicillin-containing TB medium and grown until an OD_600_ of 1 was reached. The addition of 100 µM of isopropyl-B-D-thiogalactoside induced Cyt*c* expression. Cell pellets were collected after 8 h of expression via centrifugation at 1900 RPM for 55 min at 4 °C (Shovell SS-34, Thermo Fischer Scientific, Waltham, MA, USA). Bacterial pellets were harvested and lysed, and Cyt*c* was purified using modified ion exchange chromatography; a DE52 anion-exchange column (20 mM phosphate buffer, pH 7.5, 4.0 mS/cm conductivity) followed by CM52 cation-exchange (30 mM phosphate buffer, pH 6.5, 6.0 mS/cm conductivity). Next, the protein was eluted with a high salt elution buffer (40 mM phosphate buffer, 0.5 M NaCl). Finally, the protein was desalted and concentrated by centrifugation using Amicon Ultra-15 3 kDa centrifugal filter units (Millipore; Billerica, MA, USA) and stored at −80 °C [48]. Protein concentration was determined spectrophotometrically, as detailed below. Protein purity was determined via Coomassie blue staining of a 10% tris-tricine sodium dodecyl sulfate-polyacrylamide gel electrophoresis (SDS-PAGE) gel loaded with 500 ng of each Cyt*c* variant and ran as described below.

### 4.2. Concentration Determination of Cytc

The concentration of Cyt*c* was measured using a Jasco V-570 double beam spectrophotometer (2 nm bandwidth, 200 nm/min scanning speed) using a 0.1 mm path length quartz cuvette. The Cyt*c* variants were diluted and reduced with sodium dithionite (Na_2_S_2_O_4_) or oxidized with potassium ferricyanide (K_3_Fe(CN)_6_). Cyt*c* was desalted using Sephadex G-25 NAP5 columns (GE Healthcare, Chicago, IL, USA). Cyt*c* concentration was calculated using differential spectra at 550 nm by subtracting the oxidized form from the reduced form using the formula: [Cyt*c*] = (A_550_reduced − A_550_oxidized)/(19.6 mM/cm × 1 cm) × dilution factor. The presented reduced UV–Vis spectra were recorded at 25 μM of Cyt*c*, while the oxidized UV–Vis spectra were recorded at 200 μM of Cyt*c*.

### 4.3. Cytochrome c Oxidase Activity Measurement

Regulatory-competent COX was previously purified from bovine heart tissue as described [50] and diluted to 3 µM in solubilization buffer (10 mM of K-HEPES, 40 mM of KCl, 1% Tween 20, 2 mM of EGTA, 10 mM of KF, pH 7.4 supplemented with a 40-fold molar excess of cardiolipin and 0.2 mM of ATP). Prior to use, COX was dialyzed overnight in 1 L of solubilization buffer using a 12–14 kD Spectra/Por 2 dialysis membrane (Spectrum Laboratories, Inc.; Rancho Dominguez, CA, USA) at 4 °C to remove cholate bound to the COX during the purification process. COX activity was measured in a Clark-type oxygen electrode (Oxygraph system, Hansatech; Pentney, UK) at a final concentration of 40 nM of bovine heart COX in the chamber at 25 °C in 220 µL of solubilization with 20 mM of ascorbate as an electron donor. Oxygen consumption of COX was measured upon titration of Cyt*c* variants injected into the airtight oxygen electrode at different concentrations ranging from 0 to 25 µM. COX activity is reported as turnover number (s^−1^).

### 4.4. Measurement of Caspase-3 Activity

Caspase-3 activation was measured using a cell-free apoptosis detection system. Cyt*c*^−/−^ mouse embryonic fibroblasts (CRL 2613, ATCC; Manassas, VA, USA) were cultured in 150 mm dishes, pelleted, and washed with ice-cold phosphate-buffered saline (PBS) twice, followed by one wash with cytosol extraction buffer. The cell suspension was transferred to a 2 mL Dounce homogenizer with CEB buffer (20 mM K-HEPES, 10 mM KCl, 1.5 mM MgCl_2_, 1 mM EDTA, 1 mM EGTA, 1 mM dithiothreitol, 1 mM PMSF, pH 7.5) for 15 min at 4 °C to allow the cells to swell. The cells were ruptured using a pestle for 20–30 strokes. The lysates were centrifuged at 15,000× *g* for 15 min at 4 °C to remove cellular debris. The protein concentration of the supernatant was measured using a DC assay kit (Bio-Rad, Hercules, CA, USA). The Enzchek caspase-3 assay kit 2 (Thermo Fisher Scientific) was used which contains Rhodamine 110 linked DEVD tetrapeptide, an artificial caspase-3 substrate that fluoresces after cleavage by caspase-3. A 1 mg/mL protein concentration of cell extracts was incubated with mutant Cyt*c* variants at 20 µg/mL for 2 h at 37 °C. Fluorescence was detected using a Fluorskan Ascent FL plate reader (Lab Systems, Thermo Scientific; Waltham, MA, USA) with an excitation filter of 485 nm with 14 nm bandwidth and an emission filter of 527 nm with 10 nm bandwidth. Readings were corrected using background measurement in the absence of Cyt*c*. The final reading is expressed in the percentage of change compared to WT.

### 4.5. Measurement of Cytochrome c Oxidation and Reduction Rate

The kinetics of oxidation of ferro-Cyt*c* by H_2_O_2_ were measured spectrophotometrically at 550 nm [68]. Cyt*c* variants were fully reduced with sodium dithionite (Na_2_S_2_O_4_) and desalted using NAP5 columns (GE Healthcare, Piscataway, NJ, USA). The initial absorbance of 15 µM Cyt*c* in 0.2 M Tris-Cl, pH 7.0, was measured at 550 nm and 630 nm as a background reading using 0.2 M Tris-Cl as a blank. Then, 100 μM of H_2_O_2_ was added to the cuvette as an oxidizing agent, and the reduction in absorbance at 550 nm was measured every 10 s for 1 min. The rate of Cyt*c* oxidation is expressed in μM/s.

The kinetics of reduction of ferri-Cyt*c* by superoxide were measured spectrophotometrically at 550 nm. The rate of reduction was measured using a hypoxanthine/xanthine oxidase system to generate superoxide [69]. Cyt*c* variants were fully oxidized with potassium ferricyanide (K_3_Fe(CN)_6_) and desalted using NAP5 columns (GE Healthcare, Piscataway, NJ, USA). A reaction mixture consisting of 10 µM of Cyt*c*, 100 μM of hypoxanthine, and 14.2 nM of catalase in 1×PBS was prepared in a cuvette, and an initial absorbance was measured at 550 nm using 1×PBS as a blank. The reaction was initiated with the addition of 181.5 nM xanthine oxidase to the cuvette. The increase in absorbance at 550 nm was measured every 15 s for 3 min. Superoxide dismutase 2 (925 nM) was used as a negative control by detoxifying superoxide. The rate of Cyt*c* reduction is expressed in μM/s.

### 4.6. Heme Degradation Assay

The degradation of the covalently attached heme group of Cyt*c* using high concentration H_2_O_2_ was measured spectrophotometrically. Cyt*c* variants were oxidized using potassium ferricyanide (K_3_Fe(CN)_6_) and desalted using NAP5 columns (GE Healthcare, Piscataway, NJ, USA). A reaction mixture containing 5 μM of Cyt*c* in 50 mM of phosphate buffer with pH 6.1 was prepared, and the initial absorbance at 408 nm, which corresponds to the characteristic heme Soret band, was measured [68]. Heme degradation was initiated with the addition of 3 mM of H_2_O_2_. The loss in absorbance at 408 nm was measured at 60, 200, 400, 600, and 800 s. The heme degradation is reported as a percent change in absorbance compared to the baseline absorbance.

### 4.7. Cell Culture and Stable Transfection of Cytc Constructs

The codon corresponding to Y97 of the somatic rodent Cyt*c* (WT) cDNA cloned into the pBABE-puro expression plasmid (Addgene, Cambridge, MA, USA) was mutated to a glutamate residue (Y97E) as a phosphomimetic replacement or phenylalanine as an additional control (Y97F) which cannot be phosphorylated using site-directed mutagenesis (Agilent Technologies, Santa Clara, CA, USA). The same mutagenesis primers as above were used to generate the variant Cyt*c* expression constructs, which were stably transfected into Cyt*c* double knockout mouse lung fibroblasts (a kind gift from Dr. Carlos Moraes, University of Miami, Coral Gables, FL, USA) using Transfast transfection reagent (Promega, Madison, WI, USA) in a 1:1 transfection reagent to DNA ratio, as described in the manufacturer’s protocol. As a negative control, cells were also transfected with a pBABE plasmid that did not contain the sequence for Cyt*c*, resulting in an empty vector (EV) cell line. The transfected cells were cultured at 37 °C with 5% CO_2_ in DMEM supplemented with 10% fetal bovine serum (FBS) (Sigma-Aldrich, St. Louis, MO, USA), 100 µg/mL primocin (Invivogen, San Diego, CA, USA), 1 mM sodium pyruvate, and 50 µg/mL uridine. Initial selection was performed with the above media supplemented with 2 µg/mL puromycin.

### 4.8. Gel Electrophoresis and Western Blotting

Cells expressing Cyt*c* variants were lysed using 100 μL of a RIPA lysis buffer (150 mM of NaCl, 5 mM of EDTA, pH 8.0, 50 mMTris, pH 8.0, 1% NP-40, 0.5% sodium deoxycholate, 0.1% SDS) supplemented with protease inhibitor cocktail (#P8340, MilliporeSigma, Burlington, MA, USA) to prevent lysate degradation. Mixtures were then sonicated and centrifuged at 16,900× *g* for 20 min at 4 °C to remove cellular debris. SDS-PAGE of resulting lysates was carried out using 50 µg total protein of each cell line on a 10% tris-tricine gel in the presence of anode (200 mM Tris, pH 8.9) and cathode buffers (100 mM of Tris, 100 mM of Tricine, 0.1% SDS, pH 8.25). The gel was transferred onto a PVDF membrane (Bio-Rad) using a Trans-Blot SD semi-dry apparatus (#1703940; Bio-Rad) at 75 mA for 15 min and blocked in 5% milk for 1 h at room temperature. Membranes were cut between 25 and 35 kDa ladder markers. The lower and upper membranes were incubated in 1:4000 mouse anti-Cyt*c* (#556433; BD Pharmingen; San Jose, CA, USA) or 1:8000 mouse anti-GAPDH (#60004-1-Ig; Proteintech; Rosemont, IL, USA) respectively, in 5% milk overnight at 4 °C. The next day, membranes were incubated in 1:10,000 sheep anti-mouse IgG conjugated to horseradish peroxidase secondary antibody (#NA931V; GE Healthcare; Chicago, IL, USA) in 5% milk for 2 h at room temperature. The blots were visualized using Pierce ECL western blotting substrate (#32106; Thermo Fisher Scientific; Waltham, MA, USA).

### 4.9. Mitochondrial Stress Test

The OCR and extracellular acidification rates (ECAR) of the cells expressing the Cyt*c* variants were measured. Cells were seeded at a density of 10,000 cells/well in a 0.1% gelatin-coated XF24 plate (Agilent, #100777-004) overnight in 250 µL/well growth media. After overnight incubation, the growth medium was replaced with 675 µL of Seahorse media was prepared with 4.15 g of DMEM powder (#D5030, Millipore Sigma) dissolved in 500 mL of ddH_2_O, pH 7.4, sterile filtered (#431097, Corning Incorporated, Corning, NY, USA), supplemented with 10 mM glucose and 10 mM sodium pyruvate without FBS or phenol red. The cells were incubated in a CO_2_-free incubator for 1 h with seahorse media. Then, the OCR and ECAR were measured in an XFe24 Seahorse extracellular flux analyzer (Seahorse Biosciences, North Billerica, MA, USA). Following basal measurements, sequential injections of oligomycin (1 μM), carbonyl cyanide chlorophenylhydrazone (CCCP, 2.5 μM), and rotenone/antimycin A (1 μM) were performed. The mitochondrial stress test parameters (basal respiration, non-mitochondrial respiration, proton leak, ATP-coupled respiration, maximal respiration, and spare respiratory capacity) were calculated according to the manufacturer’s instructions. OCR is reported as pmol O_2_/min, and ECAR is reported as mpH/min.

### 4.10. Measurement of Membrane Potential

Mitochondrial membrane potential (ΔΨm) was measured using the JC-10 (5,5′,6,6′-tetrachloro-1,1′,3,3′-tetraethylbenzimidazolylcarbocyanine iodide) probe (#ENZ-52305, Enzo Life Sciences; Farmingdale, NY, USA). When the membrane potential is low, the JC-10 probe occurs as monomers and emits green fluorescence. When the membrane potential is high, the JC-10 probe aggregates inside the mitochondria, producing red fluorescence. Measuring this ratio between red and green fluorescence provides a relative measure of the mitochondrial membrane potential. Cells were seeded at a density of 10,000 cells/well in a black 96-well plate (Corning, #CLS3603) coated with 0.1% gelatin. The growth medium was exchanged with FBS-free, phenol red-free DMEM cell culture medium supplemented with 3.0 μM of JC-10 and cultured for 30 min. Cells were washed with 1×PBS twice. Green fluorescence (excitation 485 nm/emission 527 nm) and red fluorescence (excitation 485 nm/emission 590 nm) of the cells were measured using the Synergy H1 microplate reader (BioTek Instruments Inc.; Winooski, VT, USA). ΔΨm is represented as a ratio of red/green fluorescence.

### 4.11. Measurement of Mitochondrial ROS Production

Mitochondrial ROS production was measured using a MitoSOX red mitochondrial superoxide indicator probe (Invitrogen, #M36008). This mitochondrial ROS indicator accumulates inside the mitochondria due to its delocalized positive charge and emits red fluorescence after superoxide oxidation. Cells were seeded at a density of 10,000 cells/well in a black 96-well plate coated with 0.1% gelatin. The growth medium was exchanged with FBS-free, phenol red-free DMEM cell culture medium supplemented with 5 μM of MitoSOX and cultured for 30 min. Cells were washed with 1×PBS. Fluorescence (excitation 510 nm/emission 580 nm) was measured using a Synergy H1 plate reader (BioTek, Winooski, VT, USA). Data is reported as a percentage change compared to WT.

### 4.12. Measurement of Cell Death Using Annexin v/PI Staining

Cells expressing the Cyt*c* variants were seeded at a density of 1 × 10^6^ in 10 cm culture dishes. Cells were cultured overnight as described above in growth media. After 24 h, the cells were treated with H_2_O_2_ (400 μM for 16 h) or staurosporine (1 μM for 3 h). The dead cells were harvested by collecting the culture medium. Live cells were collected using trypsinization. The cell suspension was washed twice with 1×PBS, and 1 × 10^6^ cells were resuspended in a 1 mL 1× binding buffer (BD Pharmingen, Lot: 1026022). A total of 450 µL cell suspension was incubated with 6 µL of annexin V-FTIC (BD Pharmingen, Lot: 1026022) and 6 µL of propidium iodide (BD Pharmingen, Lot: 1026022) at room temperature in the dark for 15 min. After incubation, reagents were diluted with 3 mL of 1× binding buffer. Cells were counted on a CyFlow Space flow cytometer (Sysmex America, Inc.; Lincolnshire, IL, USA) and analyzed using FCS Express 7 software (De Novo Software; Glendale, CA, USA).

### 4.13. Oxygen Glucose Deprivation/Reoxygenation Model

Cell lines expressing Cyt*c* variants were studied using an OGD/R model to simulate I/R injury. Cells were seeded in 96 black well plates with clear bottoms and cultured to our standard protocol, as described above. Before the experiment JC-10 or MitoSOX experiment, the media was exchanged with glucose-free, FBS-free, phenol-red-free DMEM (Gibco, #A1443001) that had been equilibrated with 95% N^2^ and 5% CO^2^, which mimics the ischemic condition [49]. The cells were incubated with this ischemic media for 90 min at 37 °C with 1% O_2_ and 5% CO_2_ gas in a hypoxic chamber under the control of ProOx 110 oxygen and ProCO_2_ 120 carbon dioxide probes (Biospehrix, Redfield, NY, USA). The ischemic media were exchanged with glucose, FBS-free, phenol red-free DMEM culture medium (Gibco, Waltham, MA, USA, #31053028) and reoxygenated for 30 min under normal conditions along with the respective probe (JC-10 or MitoSOX). The control plates were maintained in normoxic conditions with regular media supplemented with FBS. The JC-10 or MitoSOX experiments were then carried out as described above.

### 4.14. Statistical Analyses

The data shown represent the mean. Error bars represent the standard deviation. For most assays, statistical analyses for the data were analyzed for statistical significance using one-way ANOVA comparing the mean of each column with the mean of every other column followed by post-hoc Tukey test using GraphPad version 10.4.0 (GraphPad Software, San Diego, CA, USA). For COX activity, the Km and Vmax for WT and Y97F were calculated using nonlinear regression followed by Michaelis-Menten calculation, while the Km (Khalf) and Vmax for Y97E were calculated using nonlinear regression followed by allosteric sigmoidal calculation. The statistical significance was calculated using one-way ANOVA with post-hoc Tukey test as described above, specifically on the 25 uM of Cyt*c* condition. For heme degradation, statistical significance was calculated using one-way ANOVA with post-hoc Tukey test as described above, specifically on the 800 s condition. For annexin V/propidium iodide experiments, statistical significance was calculated using one-way ANOVA with post-hoc Tukey test as described above on total cell death (PI+ cells, annexin V+ cells, and annexin V+/PI+ cells combined). For OGD/R experiments, statistical significance within the hypoxia and normoxia groups was calculated using one-way ANOVA comparing the mean of each column with the mean of the control column (WT) with the post-hoc Dunnett test. However, a student’s two-tailed *t*-test assuming equal variance was used to compare the hypoxia and normoxia conditions for a specific Cyt*c* variant. *p* values are indicated in the figures.

## Figures and Tables

**Figure 1 ijms-26-01314-f001:**
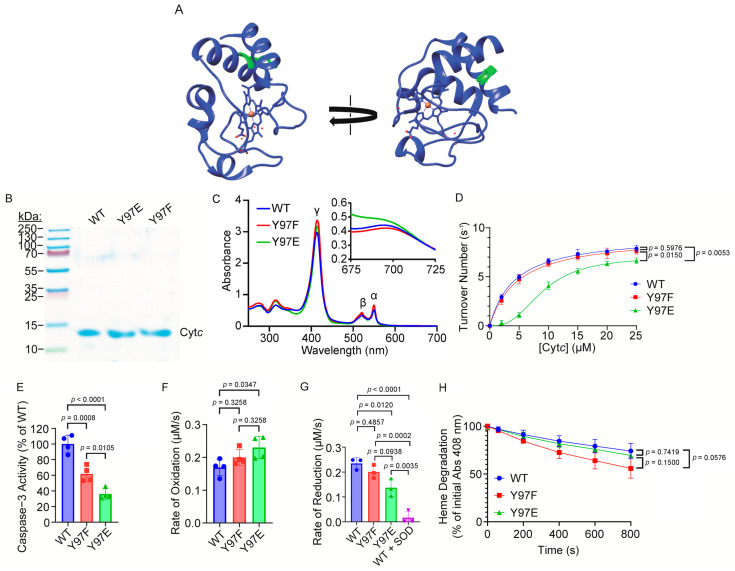
(**A**) Crystal structure (5C0Z.pdb) of rodent WT Cyt*c* (blue) [48] with Y97 (green) labeled. The PDB file was processed with the program Chimera (version 1.16). (**B**) Representative Coomassie Blue stained 10% tris-tricine SDS-PAGE gel showing purity of the recombinant Cyt*c* variants. (**C**) Reduced Cyt*c* UV–Vis spectra and oxidized Cyt*c* UV–Vis spectra (inset) indicating correct folding of the recombinant Cyt*c* variants with characteristic α, β, and γ peaks labeled. (**D**) The oxygen consumption rate of bovine heart COX in reaction with the recombinant Cyt*c* variants was measured using an Oxygraph+ system at 25 °C (*n* = 3). (**E**) Downstream caspase-3 activity in reaction with the recombinant Cyt*c* variants was measured using rhodamine fluorescence resulting from caspase-3 mediated cleavage of Z-DEVD-R110 (*n* = 4). (**F**) Initial rate of oxidation of reduced recombinant Cyt*c* variants by 100 µM of H_2_O_2_ (*n* = 4). (**G**) Initial rate of reduction of oxidized recombinant Cyt*c* variants by superoxide generated with a hypoxanthine/xanthine oxidase system (*n* = 3). (**H**) Heme degradation of recombinant Cyt*c* variants by 3 mM of H_2_O_2_ (*n* = 3).

**Figure 2 ijms-26-01314-f002:**
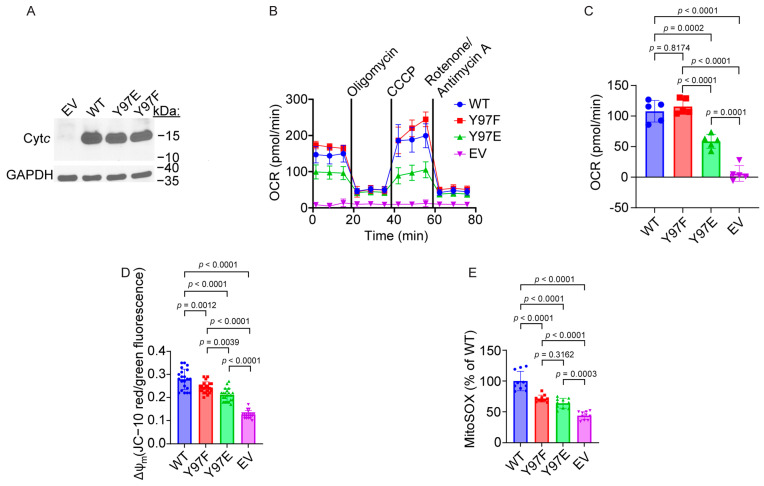
(**A**) Representative Western blot of Cyt*c* double knockout cells transfected with Cyt*c* variants, showing equal Cyt*c* expression and GAPDH as a loading control. (**B**) Mitochondrial stress test performed with sequential injections of 1 μM of oligomycin, 2.5 μM of carbonylcyanide-3-chlorophenylhydrazone, and 1 μM of rotenone/antimycin A (*n* = 5). (**C**) Basal oxygen consumption rate (OCR) from mitochondrial stress test (*n* = 5). (**D**) Mitochondrial membrane potential (ΔΨm) measured using a JC-10 probe (*n* = 21). (**E**) Mitochondrial ROS production measured using a MitoSOX probe (*n* = 10).

**Figure 3 ijms-26-01314-f003:**
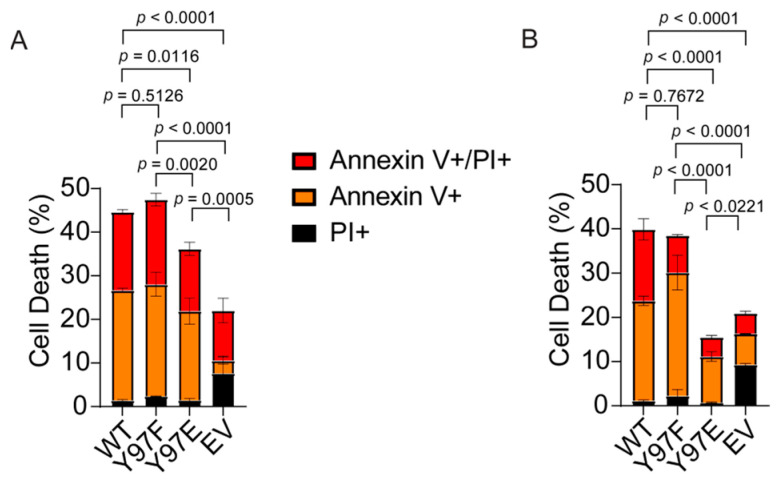
(**A**) Annexin V/propidium iodide flow cytometry data after exposure to 400 μM of H_2_O_2_ for 16 h (*n* = 3). (**B**) Annexin V/propidium iodide flow cytometry data after exposure to 1 μM of staurosporine for 5 h (*n* = 3).

**Figure 4 ijms-26-01314-f004:**
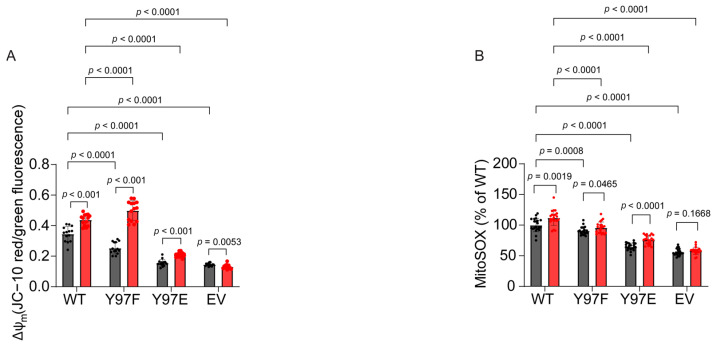
(**A**,**B**) ΔΨm (*n* = 14) and mitochondrial ROS production (*n* = 21) were measured at normoxia (gray bars) or after 90 min of oxygen–glucose deprivation followed by 30 min of reoxygenation (red bars).

**Figure 5 ijms-26-01314-f005:**
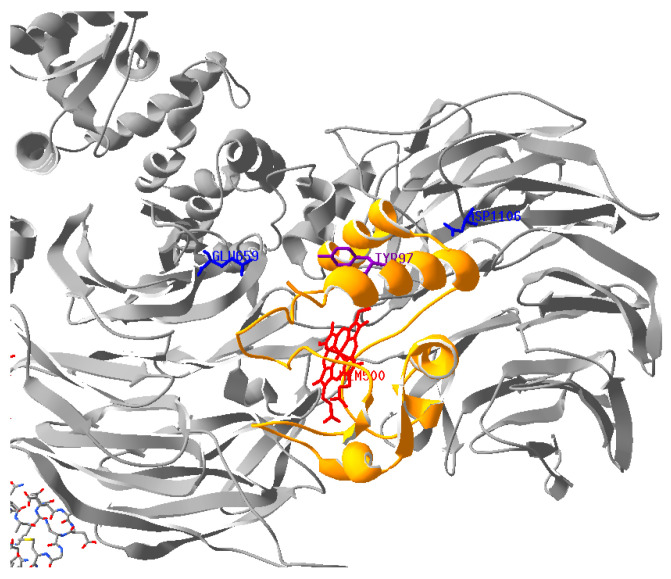
Docking model of Cyt*c* and Apaf-1 (3JBT.pdb) [62]. The distance between the amino acid residues E659 and D1106 (blue sticks) of Apaf-1 (gray) to Y97 (purple) of Cyt*c* (yellow with red heme group) is approximately 14 Å. The PDB file was processed with the program Swiss PDB viewer (version 4.1.0).

**Figure 6 ijms-26-01314-f006:**
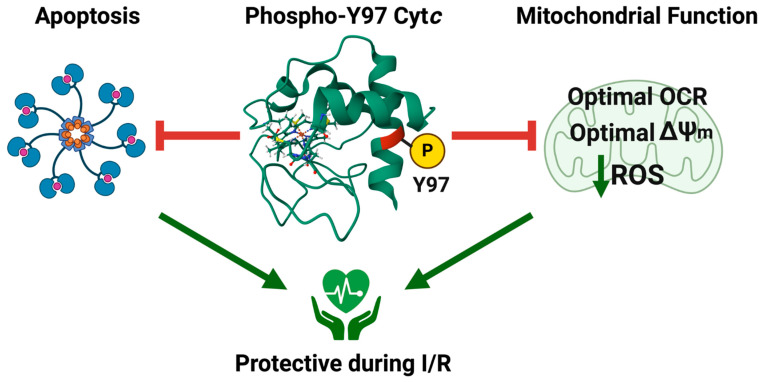
Model of Cyt*c* Y97 phosphorylation as protective to cardiac muscle.

## Data Availability

The original contributions presented in this study are included in the article/Appendix A. Further inquiries can be directed to the corresponding author.

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
