# Peer review of "Cardiac Tyrosine 97 Phosphorylation of Cytochrome *c* Regulates Respiration and Apoptosis"

_ijms, 2025, doi:10.3390/ijms26031314_

Round 1
Reviewer 1 Report
Comments and Suggestions for Authors
In this manuscript the authors use a phophomimetic Y97E variant of rodent cytochrome c (cytc) to study the effects of phosphorylation at this site on the electron transport and apoptotic function of cytc. A Y97F variant (cannot be phosphorylated) is used as a control. Both in vitro and in vivo experiments to determine the effects of phosphorylation at this site on function are presented. The results rather convincingly show that phosphorylation at this site downregulates the electron transport function of cytc to an optimal level to limit production of ROS and that it also protects against apoptosis. These results are consistent with previous work from this lab on cow cytc phosphorylated at this position. A few suggestions follow.
1. At the beginning of Results (section 2.1), be specific about the source of the cytc (i.e. rodent) and perhaps note that mouse and rat cytc have the same primary structure.
2. In Figure 2B, label the a, b and g absorption bands.
3. At the end of section 2.2, lines 116 – 117, it would be good to note that the previous work with cytc phosphorylated at position 97 was done with cytc from cow and provide an in text citation to ref 24.
4. Since Y97F is being used as a control, it would be useful to provide p values between Y97F and Y97E in Figures 1D, 1E, and 1F and in Figures 2C, 2D and 2E. Also more comparisons between the effects of the Y97F variant versus WT and the Y97E variant would be helpful for the data in Figures 1 through 4. In the current manuscript, this discussion is limited to the effect of the Y97F variant on the rate of heme degradation (Figure 1G, lines 144 – 146). Sometimes the effects are in the direction that one would expect for Y97F (cannot be phosphorylated) relative to WT (Figure 4A). But sometimes they are not. For example, phosphorylation is thought to reduce membrane potential (Y97E does this in Figure 2D), but there is also some reduction in membrane potential for Y97F, which cannot be phosphorylated (and presumably some of WT rodent cytc is phosphorylated in vivo). Also, the Y97F and Y97E variants have similarly decreased ROS production (Figure 2E) even though Y97F cannot be phosphorylated.
5. I could not find a caption for Supplementary Figure 1 – only a .tif file was provided.
6. For the pLW01 plasmid please indicate on line 356 what species the heme lyase is from.
Reviewer 2 Report
Comments and Suggestions for Authors
In the present manuscript, Hüttemann and coworkers exploited the phosphomimetic Y97E variant of rodent cytochrome c to characterize in vitro and in cell culture models the functional effects of the phosphorylation of Tyr97, observed in cow heart tissue under physiological conditions.
It turns out that Y97E Cytc features a reduced activity in the reaction with purified cytochrome c oxidase, a reduced rate of reduction, a decreased caspase-3 activity and an increased resistance to heme degradation. Moreover, the cells expressing Y97E Cytc are characterized by decreased respiration, mitochondrial membrane potential and reactive oxygen species production and are less prone to undergo apoptosis. In a cell culture model of ischemia/reperfusion injury, mitochondrial membrane potential and reactive oxygen species production were decreased. According to the authors, the result reported in the manuscript indicate that Cytc phosphorylation controls the overall electron flux along the electron transport chain by maintaining intermediate ΔΨm potentials best suited for efficient ATP production and to minimize reactive oxygen species production, thus protecting the cell from apoptosis.
The manuscript addresses a very interesting topic, given the broad interest in understanding the effect of posttranslational modifications on the different physiological roles of Cytc. Hence the scope and content of the manuscript fully match those of International Journal of Molecular Sciences.
Nevertheless, in my opinion some points need to be further clarified and addressed. Hence, the present version of the manuscript by Krantz and coworkers requires minor changes to be accepted for publication in International Journal of Molecular Sciences.
Major points
1. In my opinion a figure reporting the detailed three-dimensional structure of Cytc, clarifying the protein environment surrounding Y97 and its position with respect to the heme, would be extremely helpful for the readers to understand the structural basis of the effects exerted by Y97E on the molecular reactivity of the protein. This information is not provided by Fig. 5.
2. Page 2, Lines 66-68. The authors report that phosphorylation at Y97 shifted the charge transfer band typical of ferric Cytc from 695 nm to 687 nm, indicating perturbations of the heme environment that alter the Met80-Fe(III) bond. Is this blue shift observed in the electronic spectrum of the ferric Y97E mutant? This is an important point to understand if the structural effects of the Y97E mutation are similar to those induced by Y97 phosphorylation.
3. Page 3. Lines 121-122. Are the corresponding data for phosphorylated Y97 Cytc available? In that case, they should be cited in the text, or the corresponding reference should be added.
4. Page 9, Lines 311-313. In my opinion, the authors’ suggestion that the decreased rate of reduction of phosphomimetic Y97E Cytc by superoxide could be due to the repulsion between the negative charges of the sidechain of phosphomimetic E97 and the superoxide anion is rather vague. To be more convincing, it should be supported by a detailed analysis of the possible effect of the Y97E mutation on the structure and electrostatics of the metal site.
